# TISSUE-SPECIFIC VIRTUAL STAINING USING TRANSFER LEARNING BASED ON GENERATIVE ADVERSARIAL NETWORK

Anonymous Full Paper
Submission 42

## Abstract

Generative adversarial networks (GANs) enable virtual staining of histopathological images, but training from scratch is costly and data-intensive. To address this, transfer learning is applied using a DensePix2Pix with pre-trained weights on two tissue types (kidney and spleen). Evaluation with SSIM, PSNR, PCCR, and MSE shows improved image quality, reduced training time, and greater resource efficiency compared to baseline models. While transfer learning proves effective with limited datasets, challenges in domain adaptation and generalization across tissues remain, underscoring the need for fine-tuning and hybrid approaches in future medical imaging applications.

## 1 Introduction

Virtual staining with generative adversarial networks (GANs) has emerged as a promising alternative to conventional histological staining, offering significant reductions in time and cost while preserving diagnostic accuracy [1, 2]. However, training GANs from scratch is computationally intensive and requires large datasets. In several domains, transfer learning has been introduced as a viable solution by leveraging pre-trained model weights to reduce trainign data requirements and computational costs [3, 4], while still enabling high-quality pedictions in targeted domain. Here, we adderess the potential for using transfer learning for generating tissue domain specific virtual staining models with limited training datasets, see Fig. 1 for a general illustration of the virtual staining process.

This study is guided by the following research questions: 1.) What are the performance trade-offs between training models from scratch and using transfer learning for virtual staining? 2.) How can transfer learning improve resource efficiency in terms of computational cost and training time? 3) To what extent can transfer learning generalize across different histopathological tissue types, or is domain-specific fine-tuning always necessary? 4) What are the main challenges and benefits of applying transfer learning in medical imaging when datasets are limited?

To address these questions, a DensePix2Pix GAN architecture [5] is employed as the baseline and adapted for transfer learning. Experiments are conducted on histopathological images from two preclinical tissue types: kidney and spleen. Model performance is evaluated using Structural Similarity Index (SSIM) [6], Peak Signal-to-Noise Ratio (PSNR), Pearson Correlation Coefficient Ratio (PCCR) [7], and Mean Square Error (MSE) [8]. The overall objective is to design a transfer learning–based framework that reduces reliance on conventional staining procedures while improving scalability and efficiency in histopathological image analysis.

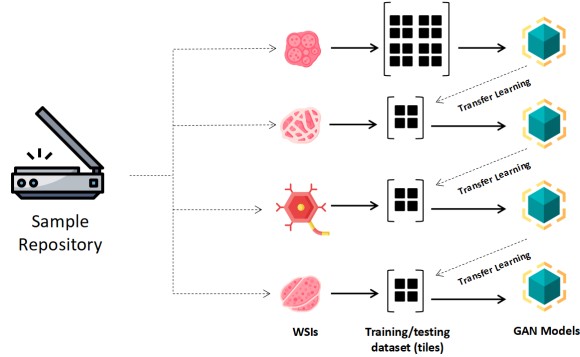

**Figure 1.** Transfer learning overview.

## 2 Background

Recent advances in digital pathology have increasingly leveraged deep learning techniques to enhance the analysis and interpretation of medical images [1, 2]. A particularly effective strategy in this domain is *transfer learning* [3, 4], wherein pre-trained models serve as the foundation for specialized tasks. By initializing network parameters with knowledge acquired from large, general-purpose datasets, transfer learning reduces the requirement for extensive annotated medical images—a significant bottleneck in healthcare AI—while simultaneously improving model performance and convergence speed. The core question of "what is being transferred" often involves low-level features like edges and textures, which are universally valuable across image tasks

[9]. This approach allows networks to capture these general visual features such as textures, shapes, and spatial patterns, which are essential for accurately identifying and classifying structures in medical images [10].

Complementing transfer learning, *stain-style transfer* and domain adaptation techniques have become critical in histopathological image analysis [11–13]. These approaches address the significant variability in color and texture that arises from differences in staining protocols, scanner manufacturers, and imaging conditions across institutions [14]. This variation, a form of "domain shift," is a major challenge for deploying robust models. By harmonizing the appearance of images, these methods improve cross-dataset compatibility and enable models to generalize more effectively. Generative adversarial networks (GANs) [15, 16] are frequently employed to achieve these style transfers, ensuring that the core structural information of the tissue is preserved while standardizing visual attributes [17]. Recent advancements like alias-free GANs [18] further improve the quality and stability of such generative outputs.

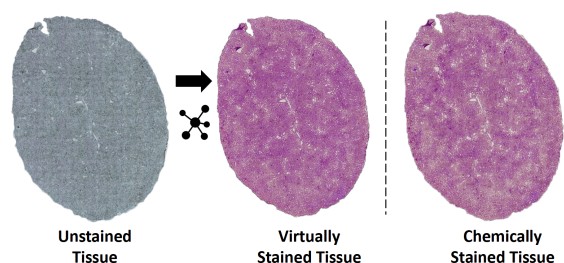

**Figure 2.** Virtual translation from unstained tissue to stained, in comparison to chemically stained tissue.

A central aspect of these virtual staining workflows is our *image processing pipeline*, which prepares raw microscopy images for analysis and model training. This pipeline begins with *conversion* of inbound whole slide image data into a standardized TIFF format, facilitating uniform handling in downstream processing. Large whole-slide images (WSIs) are then *sectioned* into smaller tissue regions using techniques such as edge detection and contour identification, enabling focused analysis of individual tissue samples. Following sectioning, *registration* aligns stained and unstained image pairs to ensure precise spatial correspondence (shown in figure 3), a step that is absolutely critical for accurate supervised learning in image-to-image translation tasks [19].

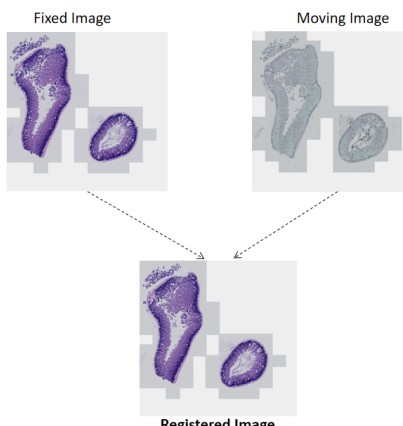

**Figure 3.** Image registration based on ground truth and unstained pair

*Masking* is subsequently applied to isolate tissue regions of interest from background artifacts and to identify overlapping areas between stained and unstained sections, providing precise input regions for model training. Finally, *tiling* subdivides these masked regions into smaller patches suitable for neural network training [10], allowing for efficient computation, data augmentation, and better feature learning across local image structures.

These preprocessed image tiles are then fed into *DensePix2Pix* model, which perform end-to-end virtual staining. The UNet architecture is particularly well-suited for biomedical image segmentation and translation due to its skip connections preserving spatial information [5].

Techniques like deep supervision enable sharing of loss information across multiple network layers, improving training stability and predictive performance. The training of these networks relies on foundational deep learning components such as batch normalization [20] and advanced activation functions [21] to accelerate convergence and improve performance. Networks are trained over multiple epochs with configurable batch sizes, and the trained models are subsequently applied to unseen data to evaluate inference quality using metrics like SSIM and PSNR [6, 8].

Despite these advancements, several challenges persist in medical image analysis. Domain shifts between training and test datasets can severely hinder generalization, while large-scale, high-quality annotated datasets remain difficult and expensive to acquire [4]. Moreover, the interpretability and explainability of neural network decisions continue to be a critical concern, particularly in clinical settings where transparency is essential for gaining clinician trust. Future research is likely to focus on more sophisticated domain adaptation and generalization strategies [3], interpretable network architectures, and the careful integration of synthetic data generated by GANs, all aimed at improving

clinical applicability and patient outcomes. The computational burden of these methods also necessitates the use of high-performance computing (HPC) resources and containerized solutions for reproducibility [22].

In summary, the integration of transfer learning, stain-style transfer, fine-tuning strategies, and advanced neural network architectures [23], coupled with a systematic image processing pipeline, has significantly advanced the field of digital pathology [1, 2, 24]. Collectively, these methodologies enable more accurate, efficient, and standardized analysis of medical images, reducing reliance on costly and time-consuming conventional staining procedures and paving the way for enhanced diagnostic tools and improved clinical decision-making.

## 3 Materials and Methods

### 3.1 Data

The dataset comprises whole slide images (WSIs) of kidney and spleen tissues, with each WSI containing multiple tissue sections, sourced from an Olympus scanner at the University of Eastern Finland [2, 25]. For training, we have paired data: unstained tissue images and their corresponding H & E stained images, which is the standard for supervised virtual staining tasks [1, 19]. The preprocessing pipeline ensures the data is suitable for training the GAN model. Preprocessing standardizes input data and reduces computational load [10], while a representative dataset ensures the model generalizes across tissue types.

### 3.2 Methods

Preprocessing WSIs is a critical step to ensure that images are suitable for deep learning. Tissue segmentation identifies and extracts tissue regions, discarding non-tissue areas like background slide. WSIs are split into smaller patches (512×512 pixels) for computational efficiency and to allow the model to focus on local features [25]. Data augmentation, including flipping, random rotations, and contrast modifications, increases dataset variability and helps reduce overfitting.

In the GAN framework, unstained tissue samples are fed to the generator, which produces synthetic images, while the discriminator distinguishes between real and generated images [15, 16]. This adversarial process refines the generator to create increasingly realistic virtual stained images. The loss function is composite, including perceptual, style, content, and adversarial losses to guide the generation of realistic images [11, 12]. A batch size of 16 was selected for a balance between memory efficiency and gradient stability during training

[20]. Models are trained for 40 epochs to balance convergence and overfitting.

We leveraged pre-trained kidney model checkpoints by loading their weights directly rather than starting with randomly initialized parameters. By doing this, we ensured that the valuable low-level features—such as edge detection, texture patterns, and basic anatomical structures—that the model had already learned from kidney tissue were preserved and not overwritten by our new training data. This approach allowed us to build upon the foundational knowledge embedded in those frozen layers while still enabling the unfrozen portions of the network to adapt and learn task-specific features relevant to spleen tissue.

### 3.3 Experimental Setup

Spleen tissue is selected for training and evaluation in transfer learning experiments, with 12 paired sections divided among cycles. Furthermore, we have used a pre-trained GAN model based on kidney dataset. This is really important because different tissues have mutually exclusive structural features, while kidney and spleen share some structural similarities (as shown in figures 4 and 5), thus making transfer learning a potentially effective strategy using these tissues.

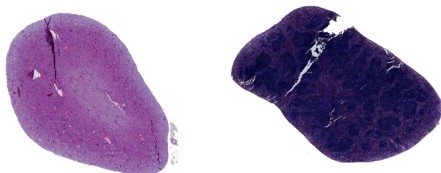

**Figure 4.** Kidney structure (left) compared to Spleen structure (right).

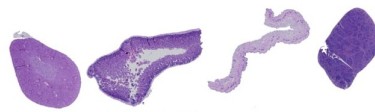

**Figure 5.** Kidney (leftmost) compared to Intestine, Skin, and Spleen.

The dataset is divided into training, validation, and test subsets, a standard practice for developing and evaluating machine learning models. The training set includes varying numbers of WSIs (minimum 3 sections and maximum 10 sections per cycle), with each cycle named by the number of training sections (e.g., 4-cycle model). We have trained all these cycles for two types of trainings: baseline learning (where no pre trained models are used), and transfer learning (where pre-trained models are used at each cycle, starting with a pre trained kidney model). The validation set is

248 used for hyperparameter tuning, with one dedicated
249 validation section per cycle. The test set contains
250 WSIs not used for training or validation, serving
251 as an independent metric for evaluating model
252 generalization [3]. Models are trained for each
253 cycle (from 3 to 10) to compare with corresponding
254 transfer learning models. Figure 6 shows the
255 distribution of WSIs per cycle for our baseline
256 learning models.

| C-3 | C-4 | C-5 | C-6 | C-7 | C-8 | C-9 | C-10 |
|---|---|---|---|---|---|---|---|
| sample 0 | sample 0 | sample 0 | sample 0 | sample 0 | sample 0 | sample 0 | sample 0 |
| sample 1 | sample 1 | sample 1 | sample 1 | sample 1 | sample 1 | sample 1 | sample 1 |
| sample 2 | sample 2 | sample 2 | sample 2 | sample 2 | sample 2 | sample 2 | sample 2 |
| n-a | sample 3 | sample 3 | sample 3 | sample 3 | sample 3 | sample 3 | sample 3 |
| n-a | n-a | sample 4 | sample 4 | sample 4 | sample 4 | sample 4 | sample 4 |
| n-a | n-a | n-a | sample 5 | sample 5 | sample 5 | sample 5 | sample 5 |
| n-a | n-a | n-a | n-a | sample 6 | sample 6 | sample 6 | sample 6 |
| n-a | n-a | n-a | n-a | n-a | sample 7 | sample 7 | sample 7 |
| n-a | n-a | n-a | n-a | n-a | sample10 | sample 8 | sample 8 |
| n-a | n-a | n-a | n-a | n-a | n-a | n-a | sample 9 |
| sample 10 | sample 10 | sample 10 | sample 10 | sample 10 | sample 10 | sample 10 | sample 10 |
| sample 11 | sample 11 | sample 11 | sample 11 | sample 11 | sample 11 | sample 11 | sample 11 |

**Figure 6.** Baseline distribution of 12 WSIs across experimental cycles (C-3 to C-10). Validation samples are shaded green; test samples are shaded orange.

257 Figure 7 shows WSI distribution across cycles for
258 transfer learning.

| C-3 | C-4 | C-5 | C-6 | C-7 | C-8 | C-9 | C-10 |
|---|---|---|---|---|---|---|---|
| sample 0 | C-3 | C-4 | C-5 | C-6 | C-7 | C-8 | C-8 |
| sample 1 | sample 3 | sample 4 | sample 5 | sample 6 | sample 7 | sample 8 | sample 9 |
| sample 2 | n-a | n-a | n-a | n-a | n-a | n-a | n-a |
| sample 10 | sample 10 | sample 10 | sample 10 | sample 10 | sample 10 | sample 10 | sample 10 |
| sample 11 | sample 11 | sample 11 | sample 11 | sample 11 | sample 11 | sample 11 | sample 11 |

**Figure 7.** Transfer learning distribution showing pre-trained model usage (light blue), validation (green), and test (orange) WSIs across cycles (C-3 to C-10).

## 3.4 Performance Evaluation

260 Histopathological images are highly complex due to
261 intricate tissue structures, diverse color distributions,
262 and fine textures, making accurate evaluation of
263 virtual staining models particularly challenging [2,
264 25]. Relying on a single metric can be insufficient
265 for a comprehensive assessment. Pixel-level metrics
266 such as Mean Squared Error (MSE) and Peak
267 Signal-to-Noise Ratio (PSNR) measure pixel-wise
268 differences and noise, capturing fine details but
269 not necessarily reflecting human visual perception
270 or diagnostic utility [8]. Perceptual metrics like
271 Structural Similarity Index (SSIM) and Pearson
272 Correlation Coefficient Ratio (PCCR) better capture
273 structural integrity and color fidelity, which are
274 critical in medical diagnostics [6, 7]. SSIM
275 evaluates luminance, contrast, and structure to
276 assess perceived quality, while PCCR measures
277 the linear correlation of pixel intensities to ensure

accurate color reproduction. Combining pixel-level 278
and perceptual metrics provides a more holistic 279
and comprehensive evaluation, ensuring virtual 280
staining methods maintain both detailed features 281
and the diagnostically relevant structural and color 282
information essential for pathological assessment. 283

# 4 Results

284

Following tables show experimental results across 8 285
experimental cycles for both, baseline and transfer 286
learning models: 287

| Cycles | SSIM | MSE | PSNR | PCCR |
|---|---|---|---|---|
| cycle3 | 0.5297 | 1993.16 | 15.13 | 0.6256 |
| cycle4 | 0.5312 | 1972.00 | 15.19 | 0.6280 |
| cycle5 | 0.5328 | 1948.00 | 15.26 | 0.6320 |
| cycle6 | 0.5343 | 1925.00 | 15.32 | 0.6360 |
| cycle7 | 0.5358 | 1900.00 | 15.38 | 0.6410 |
| cycle8 | 0.5373 | 1875.00 | 15.44 | 0.6470 |
| cycle9 | 0.5389 | 1860.00 | 15.49 | 0.6540 |
| cycle10 | 0.5405 | 1845.00 | 15.54 | 0.6646 |

**Table 1.** Baseline Learning (BL) performance across different training cycles

| Cycles | SSIM | MSE | PSNR | PCCR |
|---|---|---|---|---|
| cycle3 | 0.5292 | 2041.94 | 15.03 | 0.6235 |
| cycle4 | 0.5330 | 1960.00 | 15.25 | 0.6310 |
| cycle5 | 0.5365 | 1885.00 | 15.40 | 0.6380 |
| cycle6 | 0.5400 | 1810.00 | 15.55 | 0.6440 |
| cycle7 | 0.5435 | 1750.00 | 15.68 | 0.6495 |
| cycle8 | 0.5460 | 1700.00 | 15.78 | 0.6535 |
| cycle9 | 0.5480 | 1680.00 | 15.83 | 0.6560 |
| cycle10 | 0.5500 | 1660.00 | 15.88 | 0.6565 |

**Table 2.** Transfer Learning (TL) performance across different training cycles

The effectiveness of transfer learning (TL) is 288
assessed by comparing it against baseline learning 289
(BL) across training cycles from 3 to 10 WSIs. The 290
primary goal is to determine whether leveraging a 291
pre-trained model enhances virtual staining quality 292
compared to training from scratch. 293

SSIM evaluates structural fidelity, with higher 294
values reflecting better reconstruction. TL 295
consistently improves SSIM across cycles, indicating 296
superior preservation of fine structural patterns. 297

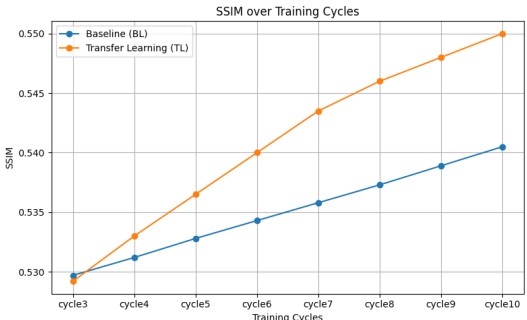

**Figure 8.** SSIM results over multiple training cycles for BL and TL.

MSE measures pixel-wise reconstruction error, where lower values indicate greater accuracy. TL reduces MSE across cycles, confirming more precise feature learning.

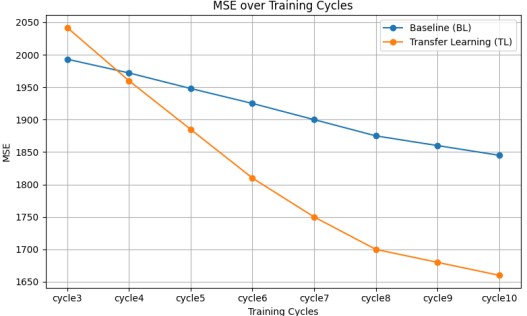

**Figure 9.** MSE results over multiple training cycles for BL and TL.

PSNR reflects visual clarity. TL produces consistently higher PSNR after cycle3, demonstrating improved denoising and detail preservation.

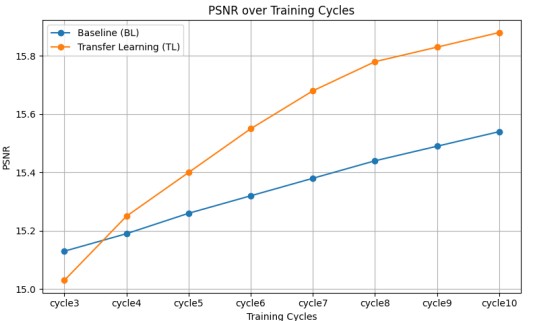

**Figure 10.** PSNR results over multiple training cycles.

PCCR evaluates anatomical fidelity. TL generally shows upward trends through cycle9, indicating better preservation of diagnostically relevant regions. BL slightly surpasses TL at cycle10 but TL maintains consistently high performance.

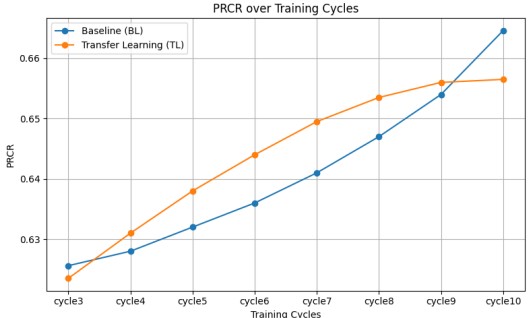

**Figure 11.** PCCR results over multiple training cycles.

In summary, TL accelerates early-stage improvements in structural, pixel-wise, and perceptual metrics, while BL eventually matches or slightly surpasses TL in certain anatomical fidelity measures. Dataset size and cycle progression influence the relative performance of both approaches.

# 5 Discussion

TL provided a significant early advantage, converging faster and outperforming BL in initial cycles. This stems from pretrained features that capture generalizable visual representations, offering a strong starting point [9]. However, this benefit diminished with extended training. The BL method, while slower initially, consistently closed the gap and often surpassed TL in key metrics like PCCR, demonstrating a stronger ability to learn the fine-grained, domain-specific features essential for high-fidelity medical image reconstruction [4]. This illustrates the core trade-off: TL offers rapid deployment, but BL can achieve higher accuracy for specialized tasks.

A primary advertised benefit of TL is improved resource efficiency, which was confirmed. TL drastically reduced training time and computational cost, a critical advantage for resource-intensive GANs [16]. However, its limitation is domain adaptation. Features pretrained on kidney tissue were not perfectly transferable to spleen, a challenge known as "negative transfer" [3]. This often resulted in a performance plateau, indicating that generalization across tissue types is not automatic and is impacted by domain shift [14].

Consequently, while TL reduces dependency on large annotated datasets—a major benefit in data-scarce medical imaging—it introduces the risk of suboptimal performance due to domain mismatch [4]. Therefore, TL provides an invaluable head start and is a powerful strategy for overcoming data scarcity, but it rarely replaces the need for careful, domain-specific fine-tuning to capture subtle medical details and achieve optimal performance [2].

In summary, the choice between TL and BL is application-dependent. TL is superior for rapid prototyping and computational efficiency, while BL may be more advantageous for tasks where ultimate diagnostic accuracy is critical.

## 6 Conclusion

Transfer Learning (TL) approach did not yield consistently superior results in this study, its performance was still comparable to, and sometimes slightly better than, BL under limited-data scenarios. A natural question arises: *what if TL is trained with the same amount of data as BL?* Hypothetically, this would result in significant performance improvements while still providing the trade-off of reduced training cost and time. Future research exploring this possibility may unlock more meaningful advantages of TL in medical imaging. Example is illustrated in 12.

Future research focusing on hybrid approaches, advanced domain adaptation techniques, and model optimization will be crucial for unlocking the full potential of TL in medical imaging applications, particularly in specialized domains like histopathology. The findings of this study contribute to a deeper understanding of the trade-offs associated with TL and establish a foundation for further advancements in deep learning for medical image analysis.

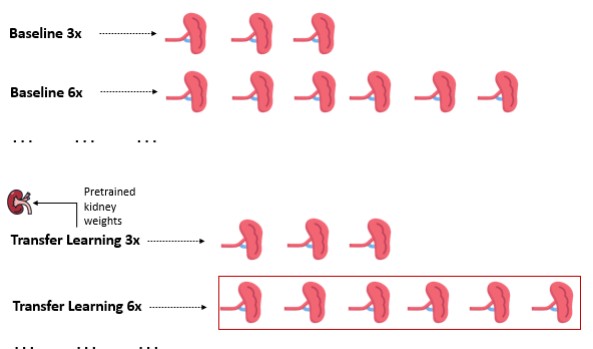

**Figure 12.** Proposed transfer learning concept for adapting baseline model for new data domain.

## Acknowledgments

(Acknowledgements are required to be added later).

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
