# OpenReview forum: "TISSUE-SPECIFIC VIRTUAL STAINING USING TRANSFER LEARNING BASED ON GENERATIVE ADVERSARIAL NETWORK"
_NLDL.org/2026/Conference — Submitted to NLDL 2026_

### Official Review · Reviewer_oa28 · 2025-09-27
**TISSUE-SPECIFIC VIRTUAL STAINING USING TRANSFER LEARNING BASED ON GENERATIVE ADVERSARIAL NETWORK**

**Rating:** 1
**Confidence:** 5

**Summary:**

This paper proposes a domain translation approach for tissue staining using transfer learning from kidney to spleen datasets to address data scarcity in medical imaging. The authors employ a GAN-based Pix2Pix framework for training. However, experimental results demonstrate limited improvements compared to baseline models.

**Strengths:**

- The method is based on a conceptually straightforward approach
- The evaluation encompasses multiple experimental settings, including cyclic training configurations

**Weaknesses:**

**Problem Formulation and Motivation:**

1. The fundamental challenges of this problem are inadequately articulated. The authors fail to clearly identify what existing methods lack and why previous approaches are data-intensive (how many images are required to produces a good result). This is particularly critical given that prior work leverages cycle consistency methods that enable unpaired domain translation, whereas this approach requires supervision. Cycle consistency-based methods also require less preprocessing (e.g., no registration), making the motivation for the proposed supervised approach unclear.

**Related Work:**

2. The literature review lacks coverage of inverse problem methods, generative prior-based approaches, and diffusion models (e.g., MCG, Deep Generative Prior). This omission limits the contextualization of the proposed work within the broader field.

**Technical Clarity and Reproducibility:**

3. The manuscript requires substantial improvement in writing quality and technical clarity. Figure captions need enhancement; for instance, Figure 1's caption lacks sufficient explanation of the transfer learning framework and the depicted content.

4. The paragraph beginning at line 212 requires revision to better explain the methodology.

5. Critical implementation details are missing, hampering reproducibility. Specifically, the authors should clarify: (1) results without fine-tuning on the target dataset, (2) which layers are frozen after pre-training (line 222).

**Experimental Design:**

6. The cycle training approach (Figure 7) lacks proper explaination. If models are initialized from previous cycles, the authors must explain why cycling is necessary rather than fine-tuning on the complete dataset. If this is not the case, the representation and purpose of these cycles require clarification.

7. Dataset details are insufficient despite provided references. The authors should specify the number of images used and the final count of 512x512 patches for pre-training and fine-tuning.

8. Normalization details (pixel value ranges) are absent, making MSE error values difficult to interpret without minimum and maximum bounds.

**Architecture and Implementation:**

9. Claims regarding training speed improvements lack supporting evidence. Training times, hardware specifications, and model parameter counts should be provided.

**Results Analysis:**

10. The authors acknowledge that baseline eventually matches or slightly surpass (line 314) the transfer learning approach with sufficient training but fail to explain this phenomenon or propose solutions, undermining the core contribution.

11. Figure 12 lacks proper explanation and detailed description.

12. Visual results showing actual stained tissue outputs from both the proposed method and baselines are absent.

**Comparative Evaluation:**

13. Given the prior application of CycleGAN-based methods to similar problems, the authors must provide comparisons or justify why such approaches are inapplicable.

## Questions

14. How does the method perform without freezing any pre-trained weights?

[MCG] Chung, Hyungjin, et al. "Improving diffusion models for inverse problems using manifold constraints." Advances in Neural Information Processing Systems (2022).

[Deep Generative Prior] Pan, Xingang, et al. "Exploiting deep generative prior for versatile image restoration and manipulation." IEEE Transactions on Pattern Analysis and Machine Intelligence (2021).

**Justification:**

The manuscript requires comprehensive revision in both technical content and presentation quality. In its current state, the methodology is difficult to understand and would be challenging to reproduce. The fundamental contribution is inadequately motivated, and the experimental validation lacks rigor. The writing quality significantly impedes comprehension.

---

### Official Review · Reviewer_CnA8 · 2025-10-06
**TISSUE-SPECIFIC VIRTUAL STAINING USING TRANSFER LEARNING BASED ON GENERATIVE ADVERSARIAL NETWORK**

**Rating:** 2
**Confidence:** 5
**Final Rating:** 2
**Final Confidence:** 5

**Summary:**

This paper explores and compares training from scratch vs transfer learning using a model trained on similar (tissue) data. It uses the same model architecture and training strategy to compare baseline models trained from scratch with increasingly many samples (WSIs) with subsequent transfer learning where one new sample is added in each cycle. The validation and test WSI is kept constant throughout the experiment. 4 difference performance metrics are used.

**Strengths:**

Interesting research questions:
1.) What are the performance trade-offs between training models from scratch and using transfer learning for virtual staining?
2.) How can transfer learning improve resource efficiency in terms of computational cost and training time?
3) To what extent can transfer learning generalize across different histopathological tissue types, or is domain-specific fine-tuning always necessary?
4) What are the main challenges and benefits of applying transfer learning in medical imaging when datasets are limited?

Good referencing to related work.

Good description of importance of similar tissue types and nice that you (try to) illustrate that in the Figures-the details can however not be seen and should also have been highlighted with eg. arrows and described in text.

Good that you use more than one performance metric.

**Weaknesses:**

General:
The small comparison done does not answer all the 4 questions listed.

The language and text is somewhat repetitive (descriptions and motivations multiple times).

The figure captions should be more descriptive (self explanatory)

How sample, WSI, and sections are related to one another is somewhat confusing from the text.

Figure 4 and 5 are very small and the spleen in Figure 4 is very dark. You should explain and highlight the differences between tissues in Figure 5.

No refs to result plot Figures.

Method:
Not enough details about the training is provided. How many layers were frozen vs unfrozen? Was 40 epochs used also for the TL (one additional sample in each cycle) vs training from scratch? If so, what does that mean for cycle 10? What hyper parameters were tuned with the validation set (and how)? How was the batch size and epochs verified?  There is no motivation to the choice of loss function and what the weights for the different parts were.

How much does which WSI is used as validation and test set influence the result? For the BL - how does a different seed affect the result or which WSIs where used for training/testing? Without any such tests on variation the values given cannot really be related to anything? Is the difference between BL and TL real or insignificant?

Discussion:
Strong conclusions made in the discussion that can not be seen or drawn based on the experiments and results e.g:
1)"The BL method, while slower initially, consistently closed the gap and often surpassed TL in key metrics like
PCCR,...." - there is no often or metrics in the plots.

2) "This illustrates the core trade-off: TL offers rapid deployment, but BL can achieve higher accuracy for
specialized tasks." -there is nothing about time or speed in the experiments.

3) "A primary advertised benefit of TL is improved resource efficiency, which was confirmed." - this is not shown/confirmed.
If your TL needs reduced training time with increasing cycles it is not so strange since you have already refined it many times in previous cycles.

4)"However, its limitation is domain adaptation. Features pretrained on kidney tissue
were not perfectly transferable to spleen, a challenge known as ”negative transfer” [3]    – This is not shown at all.

**Final Justification:**

I cannot find any replies to review comments from the authors at all.

**Justification:**

This study is not thorough enough to be a full paper. With some lesser claims and extended experiments (std for BL for example) it would be a nice extended abstract.

---

### Official Review · Reviewer_d7as · 2025-10-11
**review of  TISSUE-SPECIFIC VIRTUAL STAINING USING TRANSFER LEARNING BASED ON GENERATIVE ADVERSARIAL NETWORK**

**Rating:** 2
**Confidence:** 4

**Summary:**

The authors compare a variant of transfer learning versus training from scratch for a pix2pix GAN aimed at predicting the H&E stain for unstained tissue sections. They train from scratch and do the transfer learning training on a spleen dataset. The pretrained model was trained on kidney tissue.  They  evaluate on a spleen dataset. They compute a three independent statistical measures for the deviation between ground truth stained tissue and predicted staining.

**Strengths:**

They compare two very relevant settings for training a virtual staining GAN.

**Weaknesses:**

major:
- suboptimal transfer learning setup: see Figure 7: they freeze a snapshot, then they train it on a single WSI to obtain the next snapshot

This saves compute effort, however it does not mix data from different WSIs during one epoch inside transfer learning. In every training round, minibatches originate from only one WSI. This is more prone to overfitting and to forgetting of information from previous snapshots.

- limited experiments on the pixel-wise matching: trying out transfer on more tissues would be needed to draw conclusions. They transfer only kidney to spleen
- evaluation only on one WSI instead of cycling through different train/val/test splits

- H&E stain is usually not the final target. It is an intermediate step to make nuclei visible. pixel-wise MSEs and patch-wise SSIM are therefore of limited value. An evaluation in some meaningful downstream task between the predicted staining and the ground truth staining is missing.

- training details are largely missing: learning rates, optimizer schedules, data augmentation hyperparameters, loss function constants.
- which layers in TL are frozen ? see lines 220 ff
- dataset information is largely missing: resolution, number of different patients, squarepixels available


minor:

- PSNR is a deterministic function of MSE. That adds no information.
- Pearson correlations between ground truth and predicted staining are hard to interpret
- loss function is cited but not shown

- poor writing style using lots of LLM-style hyperboles like

Techniques like deep supervision enable sharing 131
of loss information across multiple network 132
layers, improving training stability and predictive 133
performance.

deep supervision is what  ?

The training of these networks relies 134
on __foundational__ deep learning components such as 135
batch normalization [20] and __advanced__ activation 136
functions [21] to accelerate convergence and improve 137
performance.

This sounds like LLM writing style stating well-known things:

The preprocessing182
pipeline ensures the data is suitable for training183
the GAN model. Preprocessing standardizes input184
data and reduces computational load [10], while a185
representative dataset ensures the model generalizes186
across tissue types.

Combining pixel-level 278
and perceptual metrics provides a more holistic 279
and comprehensive evaluation, ensuring virtual 280
staining methods maintain both detailed features 281
and the diagnostically relevant structural and color 282
information essential for pathological assessment.

Dataset size and cycle progression 315
influence the relative performance of both 316
approaches.

**Justification:**

See the weaknesses. The experiments are too limited. Descriptions of dataset and training are too sparse to be reproducible.

Question:
  - We know it is 40 epochs. In transfer learning one is using 40 epochs for every snapshot ?

---

### Meta-Review · Area_Chair_cCuu · 2025-11-01

**Recommendation:** Reject
**Confidence:** 4

**Metareview:**

The authors only partially engaged in the rebuttal process.

This submission explores the use of transfer learning within a GAN-based framework (DensePix2Pix) for virtual staining of histopathological whole-slide images (WSIs), aiming to reduce training costs and improve efficiency. The authors compare training from scratch with transfer learning from kidney to spleen tissue and evaluate performance using SSIM, PSNR, PCCR, and MSE.

While the topic is relevant to computational pathology and the motivation to reduce data and compute requirements is well made, the reviewers unanimously identified significant limitations in the technical depth, experimental rigour, and clarity of presentation. This includes a lack of technical detail and reproducibility, incomplete related work, and some generic unsubstantiated statements.

---

### Decision · Program_Chairs · 2025-11-05

**Decision:**

Reject

**Comment:**

Based on the reviewers and AC comments, the paper cannot be presented at the conference.